# Gene Cloning, Heterologous Expression, and In Silico Analysis of Chitinase B from *Serratia marcescens* for Biocontrol of *Spodoptera frugiperda* Larvae Infesting Maize Crops

**DOI:** 10.3390/molecules29071466

**Published:** 2024-03-26

**Authors:** Ghada M. El-Sayed, Maha T. H. Emam, Maher A. Hammad, Shaymaa H. Mahmoud

**Affiliations:** 1Microbial Genetic Department, Biotechnology Research Institute, National Research Centre, 33 El-Bohouth St. (Former El-Tahrir St.), Dokki, Cairo 12622, Egypt; 2Genetics & Cytology Department, Biotechnology Research Institute, National Research Centre, 33 El-Bohouth St. (Former El-Tahrir St.), Dokki, Cairo 12622, Egypt; maha_taimor@yahoo.com; 3Department of Plant Protection, Faculty of Agriculture, Ain Shams University, Cairo 11566, Egypt; 4Zoology Department, Faculty of Science, Menoufia University, Shibin El Kom 32511, Egypt; dr.shaymaahussein@science.menofia.edu.eg

**Keywords:** *Serratia marcescens*, *Spodoptera frugiperda*, recombinant chitinase B, cloning, expression, insecticidal activity, molecular docking

## Abstract

*Spodoptera frugiperda*, the fall armyworm (FAW), is a highly invasive polyphagous insect pest that is considered a source of severe economic losses to agricultural production. Currently, the majority of chemical insecticides pose tremendous threats to humans and animals besides insect resistance. Thus, there is an urgent need to develop new pest management strategies with more specificity, efficiency, and sustainability. Chitin-degrading enzymes, including chitinases, are promising agents which may contribute to FAW control. Chitinase-producing microorganisms are reported normally in bacteria and fungi. In the present study, *Serratia marcescens* was successfully isolated and identified from the larvae of *Spodoptera frugiperda*. The bacterial strain NRC408 displayed the highest chitinase enzyme activity of 250 units per milligram of protein. Subsequently, the chitinase gene was cloned and heterologously expressed in *E. coli* BL21 (DE3). Recombinant chitinase B was overproduced to 2.5-fold, driven by the T7 expression system. Recombinant chitinase B was evaluated for its efficacy as an insecticidal bioagent against *S. frugiperda* larvae, which induced significant alteration in subsequent developmental stages and conspicuous malformations. Additionally, our study highlights that in silico analyses of the anticipated protein encoded by the chitinase gene (ChiB) offered improved predictions for enzyme binding and catalytic activity. The effectiveness of (ChiB) against *S. frugiperda* was evaluated in laboratory and controlled field conditions. The results indicated significant mortality, disturbed development, different induced malformations, and a reduction in larval populations. Thus, the current study consequently recommends chitinase B for the first time to control FAW.

## 1. Introduction

The fall armyworm (FAW), *Spodoptera frugiperda*, is a widely adaptable pest that poses a significant threat to maize and various cereals in tropical and subtropical areas across the globe [1]. Originating from the Americas, recent incursion of the FAW into Africa, Asia, and Oceania has resulted in substantial reductions in yields of multiple food crops [2]. In May 2019, the Agricultural Pesticide Committee (APC) of Egypt reported the first case of the fall armyworm pest in Kom Ombo City, Aswan Governorate, affecting maize yield [3]. Maize (*Zea mays* L.) is recognized as a commercial crop in Egypt, with versatile uses as fuel, food, poultry feed, and fodder. Various strategies have been implemented to combat *S. frugiperda*, encompassing chemical approaches, biological control agents, and physical methods aimed to minimize crop yield losses [4]. The invasive and destructive nature of FAW has prompted a focus on applied pest management practices to ensure sustainable crop production [5]. Particularly in invaded regions like Africa and Asia, chemical methods continue to be a primary recourse for emergency FAW control, aimed at mitigating crop damage and preventing further spread [6]. However, challenges persist with chemical control methods, primarily stemming from the extensive use of conventional chemical insecticides such as cypermethrin, lambda-cyhalothrin, and chlorpyrifos [7]. This overreliance has led to the development of resistance in FAW populations, additionally posing risks such as harm to non-target organisms, environmental pollution, and threats to human health [8]. Hence, it is imperative to create and implement safer alternatives, such as the application of biological control methods, in regions affected by the invasion of the fall armyworm (FAW). A prudent approach involves isolating entomopathogenic bacteria from infected insects and cultivating them to evaluate their insecticidal impact, either through cell cultures or the examination of potential secondary metabolites, such as toxins or enzymes. An example of employing bacterial cells against *S. frugipedra* is seen in a recent investigation conducted by Chen et al. [1], who isolated entomopathogenic nematodes, specifically *Heterorhabditis bacteriophora*, demonstrating their effectiveness across laboratory, greenhouse, and field settings. Another example of utilizing secondary metabolites as insecticides is presented by Sun et al. [2], where a fungal-derived Cyclosporin A demonstrated notable insecticidal efficacy (LC50 = 9.69 μg/g) against FAWs by effectively inhibiting calcineurin (CaN) activity, resulting in significant suppression of the pests.

Chitin serves as a primary constituent in the cuticle and the peritrophic membrane, a protective covering in the digestive tracts of numerous insects [9]. This constituent is primarily degraded by enzymes called chitinase, which play a crucial role in breaking down chitin into its individual units. Various microorganisms, such as fungi like *Metarhizium anisopliae*, bacteria including *Bacillus cereus*, *Bacillus pumilus*, and *Serratia marcescens*, actinomycetes like *Streptomyces* spp., and yeasts such as *Tilletiopsis albescens* and *Tilletiopsis washingtonensis*, produce these enzymes [10]. There has been significant interest in exploring the potential of chitinase-producing microbes for the development of biopesticides. These bacteria enzymatically degrade chitin found in the insect gut’s peritrophic membrane, resulting in perforations, disease, and ultimately, death of infected larvae [11]. Recently, there has been a growing emphasis on exploring chitinase gene cloning and expression. Additionally, there is a notable interest in developing recombinant strains with innovative capabilities, making these areas prominent in chitinase research. These studies aimed to explore applications such as the creation of antifungal and insecticidal agents [12]. Computational packages are widely employed for protein sequence analyses and characterization, leveraging various structural and physicochemical properties. Computational methods present a cost-effective alternative, delivering highly accurate protein structures and a comprehensive understanding of the relationships between structure, function, and substrate–protein interactions. Among these methods, template-based modeling emerges as the most authentic and precise solution to tackle this challenge. Numerous models have been generated and evaluated using computational approaches, particularly documenting the intricate protein structures of chitinases from *S. marcescens* and various Bacillus species [13]. Consequently, a semi-rational approach becomes viable for redesigning. This knowledge has proven effective in agriculture, industry, and therapeutics, leading to significant enhancements in enzyme binding efficiency and catalytic activities [14]. In the current study, we successfully isolated and identified a strain of *Serratia marcescens*. This strain was obtained from infected *S. frugiperda* and shows promising potential for chitinase production. Subsequently, the chitinase gene was cloned and heterologously expressed, resulting in a significant production of chitinase that is being evaluated for its efficacy as an insecticidal bioagent against *S. frugiperda* larvae. Additionally, our study highlights that in silico analyses of the anticipated chitinase gene (ChiB) offered improved predictions for enzyme binding and catalytic activity.

## 2. Results

### 2.1. Molecular-Based Phylogeny of the Highest Chitinase-Producing Bacterial Isolate

The bacterial strain NRC408 recorded the highest specific activity of chitinase (250 U/mg). The 16S rRNA gene of NRC408 was efficiently amplified during the PCR reaction using the universal primer pair—8f and 1429R—resulting in a 1551 bp sequence. This nucleotide sequence was aligned with all currently available 16S rRNA sequences in the GenBank database (https://www.ncbi.nlm.nih/gov/genbank/, accessed on 10 November 2023). The alignment of the strain’s 16S rRNA gene and a subsequent database search disclosed a remarkable 99% identity with *Serratia marcescens* strain Gol3 (sequence ID: MT263018.1). Utilizing the neighbor-joining (NJ) method, phylogenetic trees were constructed, as depicted in Figure 1. The tree analysis confirmed that the NRC408 strain falls within the genus *Serratia*. Following submission to GenBank, the bacterial strain NRC408 was assigned the accession number OR793165 and defined as *Serratia marcescens* strain NRC408.

### 2.2. Gene Cloning and Heterologous Expression of ChiB 

A set of primers, Chi-F/Chi-R, successfully amplified a fragment approximately 1600 base pairs long using the genomic DNA of *Serratia marcescens* strain NRC408 as a template. The nucleotide sequence corresponds to the ChiB gene, which is recorded in EMBL as entry X15208.1. After cloning the gene into pJET1.2, the resulting vector was named pJET1.2-ChiB. Subsequently, both pJET1.2-ChiB and pET-28a (+) underwent simultaneous double digestion with EcoRI and HindIII. The pET-28a (+) vector effectively incorporated the target gene between the restriction enzyme sites, downstream of the T7 promoter, and in frame with the N-terminal region encoding 35 amino acid residues; its sequence is MGSSHHHHHHSSGLVPRGSHMASMTGGQQMGRGSEF, including a cluster of six histidine residues. The constructed vector was designated as pET-ChiB, and its proper assembly was confirmed through colony PCR, where a gene fragment of approximately 1600 base pairs was amplified using pET-F/pET-R (Figure 2A) and double digestion, resulting in 5.4 kb for pET-28a (+) and around 1600 base pairs for the ChiB gene fragment (Figure 2B). Following the transformation of the engineered vector, pET28a-ChiB, into *E. coli* BL21 (DE3), the protein expression profile was monitored and assessed under various concentrations of IPTG and incubation temperatures. Our findings underscore the crucial roles played by incubation temperature and IPTG concentration in the expression of chitinase B: the optimal IPTG concentration was 0.2 mM, and the ideal temperature was 37 °C. The expressed recombinant chitinase B was denoted as rHis6-CHiB. The specific activity of the recombinant chitinase B was enhanced to 600 U/mg, demonstrating a notable improvement over the activity observed in the mother strain.

### 2.3. Partial Protein Purification and SDS-PAGE

Chitinase B was overexpressed in *E. coli* BL21 (DE3) cells through driving by the T7 expression system. The cell pellet proteins were the major source for chitinase B extraction due to the absence of recognizable signals for secretion, as indicated by gene sequence analysis. As shown in Table 1, the partial purification of the crude recombinant chitinase enzyme was conducted through fractional precipitation utilizing ethanol. Each fraction underwent assessment for chitinase activity and total protein content. Optimal specific activity was achieved at 60–70% ethanol saturation, yielding a specific activity of 629.5 ± 1.1 U/mg. This represents a substantial 2.5-fold increase in specific activity compared to the native strain of *S. marcescens* strain 408, leading to a discernible improvement in the overall recovered enzymatic activity. The fraction primarily containing the chitinase B protein underwent electrophoresis to ascertain its molecular weight. The identified band was distinctly observed at approximately 55 kDa, which corresponds to chitinase B (see Figure 3).

### 2.4. DNA Sequence and Phylogenetic Analysis of ChiB Gene

Upon receiving and assembling the sequence, the deduced amino acid sequence was aligned with those in the UniProt database (https://www.uniprot.org/help/uniprotkb, accessed on 30 November 2023). The sequence demonstrated a remarkable identity of 98.2% with *Serratia marcescens* GenBank entry CAA33278.1. Subsequently, the nucleotide sequence was submitted to GenBank with the accession number OR885894. Homology analysis unveiled that the open reading frame (ORF) spanning 1500 base pairs commences with an ATG start codon and concludes with a TAA stop codon. This ORF encodes a pre-pro-enzyme consisting of 499 amino acids recovered at 54 kDa in molecular weight. The deduced amino acid analysis indicated that chitinase B belongs to the glycosyl hydrolase 18 family, specifically the chitinase class II subfamily, featuring the glycol 18 domain as a conserved element. The deduced amino acids preceding the N-terminal His-tag are presented in Figure 4A. SignalP analysis further identified residues 1–41 as the signal peptide but not excretion in the chitinase B protein sequence (Figure 4B), and the calculated theoretical isoelectric point (pI) was about 5.93. Subsequently, Figure 5 illustrates the phylogenetic relationship between the deduced amino acid sequence and its closest relatives obtained from the UniProt database and that verify the homology and genetic relatedness of chitinase B to that produced by *S. marcescens*. Figure 6 depicts the catalytic motif DXDXE in Glycoside hydrolase family 18 (GH18) chitinases, which comprise three acidic residues, denoted as D1, D2, and E. In the inferred amino acid sequence of chitinase B, the residues D1, D2, and E within the DXDXE motif correspond to D140, D142, and E144, respectively.

### 2.5. Homology Modeling and Structure Validation of Modeled Chitinase B

A 3D model of chitinase B was constructed using SWISS-MODEL, with the most suitable match being *S. marcescens* chitinase B (PDB ID: 1W1V; resolution: 1.85 Å). The model exhibits a remarkable 99.4% identity, a zero e-value, a GMQE value of 0.80, and a QMEAN of −1.71, indicating its reliability and high quality. A graphical representation, Figure 7, demonstrates that most residues have values close to 1, signifying good local quality estimates. Residues with values below 0.6 were deemed to be of low quality. Additionally, the modeled chitinase B structure aligns with other protein structures in the PDB, reinforcing its reliability (Figure 7B). In Figure 8, the Ramachandran plot and the accompanying statistics reveal that 93.3% of the modeled chitinase B residues are in the most favored regions, and 6.6% lie in additional allowed regions, with none in generously allowed regions and only 0.1% located in disallowed regions. This further validates the model’s high quality. The Verify3D plot (see Figure 8C) for structure validation yielded a PASS, and the 3D environment profile indicates that 97.58% of the residues have an averaged 3D-1D score ≥0.1, affirming the model’s validity. ProSA-web (https://prosa.services.came.sbg.ac.at/prosa.php) analysis produced a Z-score of −10.86, pointing to the predicted model’s exceptional quality (Figure 8D). The model attests that the protein backbone dihedral angles phi (φ) and psi (ψ) are positioned accurately in the 3D model. Moreover, ERRAT assesses the statistics of non-bonded interactions among various atom types. It graphically represents the error function values against the position of a nine-residue sliding window, computed through a comparison with statistics derived from extensively refined structures. This analysis discloses an overall quality factor of 94.467.

#### 2.5.1. Alignment of the Chitinase B Model and Template (1W1V) Structure

Figure 9A,B display the 3D representations of chitinase B and the template (1W1V). When the alignment was calculated using PyMOL Molecular Viewer (Educational-use-only PyMOL), an RMSD value of 0.084 (970 to 970 atoms) was observed, suggesting that the structures were closely connected (Figure 9C). The yellow helices represent the template structure, and the green helices represent the protein homology model. The molecules that make up the target and template are homodimers, meaning that they each have two identical chains, chains A and B. The alignment between chains A in the two molecules demonstrated that chains A in the protein model’s structure and the template structure (1W1V) matched.

#### 2.5.2. Docking and Molecular Interaction Studies

As shown in Table 2 and Figure 10, the ligand, chitin, forms two hydrogen bonds through its two hydroxyl groups inside the binding pocket of the chitinase B protein. Both hydrogen bonds act as hydrogen bond donors (HBDs) with the oxygen of Glu 144, having bond lengths of 2.0 Å and 2.1 Å and an angle of 150.90°, with an interaction affinity score of −5.71 kcal/mol. When visualizing the interaction with Chimera, four hydrogen bonds were observed. However, hydrogen bonds with lengths greater than 2.5 Å were not considered due to their weakness and their angles of 106° that cannot be considered, as the optimum angle is between 120° and 180°. Table 3 illustrates the formation of hydrogen bonds between chitin as a substrate with the S. marcescens chitinase B active site. 

#### 2.5.3. Insecticidal Efficiency of Recombinant Chitinase B against *Spodoptera frugiperda* Larvae

##### Laboratory Assessment Results

Recombinant chitinase B induced 92.75% mortality and significantly prolonged the pupal period (16.31) compared to 11.35 and 13.45 days in the control and emamectin benzoate groups (Table 4). Three days post-treatment, *S. frugiperda* larvae began to display decreased appetite when compared to the untreated larvae, and a significant reduction in larval weight was observed. The pupation rate, including both eclosion and hatching rates, was significantly influenced following larval treatments. Recombinant chitinase B treatment significantly delayed the pupal period and increased adult longevity. Treatment with chitinase B induced larval, pupal, and adult malformations (Figure 11). Larval deformities included small size and distortions. Larval–pupal intermediates were also observed. Pupal malformations included the presence of small-sized pupae with degenerated appendages detached from the integument. However, adult malformations included folding and crumbled wings as well as small size. 

### 2.6. Field Evaluation

The effectiveness of chitinase B against *S. frugiperda* after spraying in the El-kilo 56 district, Behera Governorate, is illustrated in Figure 12. The obtained results indicated that a significant reduction was obtained following treatment. The highest percentage reductions in larval pupation were achieved at 7- and 10-days post-treatment, by 75 and 88.66%, respectively, compared to 86 and 91.3% in the emamectin benzoate-treated group at 7 and 10 days post-treatment, respectively.

## 3. Discussion

This study encompassed two primary inquiries: firstly, the production and refinement of a recombinant chitinase B enzyme derived from the *S. marcescens* strain 408 by heterologous expression; secondly, the utilization and evaluation of chitinase B as a sole single bioagent in biocontrol of *S. frugipedra*. The potential insect pathogen bacterium *S. marcescens* is renowned for its efficiency in producing various hydrolytic enzymes, including chitinases, which are pivotal for their pathogenicity toward insects. An early application of chitinases in biotechnology focused on their role in biocontrol of plant pathogens [15]. In our study, we isolated the bacterial strain *S. marcescens* from infected *S. frugipedra* larvae, observing its prevalence in the culture media released by the infected larvae. The identification of this isolate involved isolating and sequencing a fragment of the genes of 16S rRNA, known for their variable and highly conserved regions. Widely used for assessing genetic diversity in biological populations and microbial classification, the sequencing of the 16S rRNA fragment is a robust approach for such purposes. The sequence analysis of the 16S rDNA gene, facilitated by various primers, has significantly enhanced our understanding of prokaryotic biodiversity. It is essential to note that recombination events among species in different regions of the 16S rRNA gene may introduce inconsistencies in phylogenies. This underscores the importance of employing longer sequences for a more accurate classification of new species, as highlighted by El-Barbary and Hal [16].

*S. marcescens* harbors three types of chitinase—chitinase A, chitinase B, and chitinase C—encoded by the genes chiA, chiB, and chiC, respectively. Subsequently, during enzyme induction in production media, *S. marcescens* was expected to produce three types of chitinases. The most effective chitinases with proven insect pathogenesis are chitinase B and chitinase C [17]. Therefore, one of the approaches by which we could address this challenge and mitigate the inherent expression limitations in the host was through heterologous expression. Nonetheless, this method poses difficulty in aligning compatible expression components with particular hosts and refining the induction conditions for expression. In this investigation, the pET28a system was utilized for heterologous expression of the gene’s open reading frame. This requires the addition of isopropyl β-D-1-thiogalactoside (IPTG), whose added concentration plays a crucial role in gene expression. In this study, different concentrations were employed; the optimum concentration was 0.2 mM at 37 °C, while at higher concentrations, we clearly observed the presence of expressed proteins, but in an inactive form, which emphasized the formation of expressed proteins as inclusion bodies. This concern was previously investigated by Ariyaei et al. [18] and Yan et al. [17]. In this research, the *E. coli* strain BL21 (DE3) expression system, in conjunction with pET28a (+), demonstrated effectiveness in producing functional chitinase B with increased yields. Likewise, previous studies have also effectively utilized this expression system for the heterologous expression of soluble and functional recombinant chitinase [12]. Other researchers have cloned and expressed *S. marcescens* chitinase genes in various bacteria, such as *Lactobacillus lactis* [19]. In a study [18], genes encoding chitinase B (chiB) and chitinase C (chiC) from *Serratia marcescens* were isolated and transformed into kurstaki and israelensis strains of *Bacillus thuringiensis*, resulting in engineered *B. thuringiensis* with higher insecticidal activity compared to parental strains against larvae of *Galleria mellonella* and adults of *Drosophila melanogaster*. 

Examining the homology of the inferred sequence of amino acids in the catalytic region of chitinase B revealed its classification within the Glycoside hydrolase family 18 (GH18) chitinases. These chitinases possess a conserved catalytic motif known as the DXDXE motif, which serves a synergistic function in the enzymatic process of chitin degradation [20,21]. The study conducted by van Aalten et al. [22] presented a detailed analysis of the substrate binding, catalytic mechanism, and product displacement of GH18 chitinases. This analysis was carried out using three SmChiB complexes. The catalytic process can be divided into three main steps: (i) Substrate binding in the cleft: The substrate-binding cleft can accommodate multiple GlcNAc units, and hydrolysis takes place between the −1 and +1 GlcNAc units. Crystal structures have provided insights into the role of various amino acids in binding the −1 sugar, particularly Asp142 in the catalytic triad DXDXE. Asp142 interacts with an adjacent aspartate (Asp140) without a substrate and remains distant from the proton donor amino acid (Glu144). However, upon substrate or inhibitor binding, Asp142 rotates toward Glu144, facilitating hydrogen bond interactions. (ii) Intermediate formation and glycosidic bond breaking: Distortion of the −1 sugar enables interaction with Glu144, which acts as a proton donor and promotes the departure of the leaving group. At the same time, the 2-acetamido moiety of the −1 sugar acts as a nucleophile, resulting in the formation of a covalent bicyclic oxazolinium-ion intermediate and the breaking of the glycosidic bond. (iii) Product displacement and ring opening via a water molecule attack: After the breakdown of the glycosidic bond, the product exits the active site, and a water molecule enters. The water molecule then undergoes a general base-catalyzed attack on the anomeric center, leading to the breakdown of the intermediate. Both cyclization and ring opening occur with an inversion of stereochemistry, resulting in the overall reaction retaining stereochemistry. 

SDS electrophoresis revealed a molecular weight of 55 kDa, consistent with theoretical calculations of the deduced protein. This result aligns with findings reported by Emruzi et al. [23]. In a study by Gal et al. [24], the molecular weight of purified chitinase B from the mother strain (*S. marcescens*) was identified as 54 kDa. The one-kilodalton increase in the molecular weight of rHis6-ChiB in our study is estimated to be due to the presence of the His tag protein upstream of the amino acids of chitinase B.

Insect genomes naturally contain a significant number of GH18 chitinases, which have important functions in the breakdown and reformation of the cuticle and peritrophic matrix during molting and development [25]. However, excessive expression and activity of chitinase can interfere with chitin synthesis even after the formation of new epidermis, potentially hindering insect development and causing mortality [26]. Likewise, the external application of chitinase can have harmful effects on the newly synthesized epidermis by causing its degradation, mainly through specific protein interactions with recently formed chitin molecules [27].

The notion of utilizing chitinases as bioagents against *S. frugipedra* stems from the recognition that chitin serves as the principal structural constituent, interfacing with diverse protein mechanisms and constituting integral elements of the insect epidermis, trachea, and peritrophic membrane [21]. Consequently, the deployment of this bioagent (chitinase B) primarily targets chitin, initiating degradation of the cell wall, thereby inducing malformation and developmental suppression in the insect. Given its crucial role in insect development and its absence in higher organisms, targeting chitin has become a safe and effective strategy in the development of new pesticides [25,28]. With this concept in mind, our study focuses on the overproduction of recombinant chitinase and the estimation of its efficiency as an insecticide. Purified chitinase has been identified for its insecticidal activity against insects other than *S. frugipedra* [29]. Interestingly, numerous investigations have highlighted the efficacy of chitinase against various pathogenic organisms. Several studies have utilized *S. marcescens* as a bioagent for controlling plant pathogens, encompassing fungi and insects, as emphasized by Kshetri et al. [30]. Aggarwal et al. [31] specifically investigated the impact of the virulent *S. marcescens* strain SEN on controlling *Spodoptera litura*. Their findings indicated that this virulent strain displayed significantly higher chitinase and protease enzyme activity, leading to a reduction in the overall growth yield of *S. litura*. In the findings of Danişmazoğlu et al. [12], it was reported that 1000 U/mL of ChiA, ChiB, and ChiC exhibited insecticidal activities of 47%, 50%, and 66% on *M. neustria*, and 80%, 45%, and 50% on *H. armigera* larvae within a 10-day period, respectively. In another study by Downing et al. [32], enhanced biocontrol of the sugarcane borer *Eldana saccharina* was achieved by co-introducing the cry1Ac7 gene from *B. thuringiensis* strain 234 and the chiA gene from *S. marcescens* into strains of *Pseudomonas fluorescens*. Insecticidal activity of three chitinases isolated from symbiotic bacteria nematodes was evaluated against *Galleria mellonela* [33]. Chitinase and endotoxins derived from *Bacillus thuringiensis* have been successfully used in the biological control of *Spodoptera exigua* and *Helicoverpa armigera* larvae. They have also been shown to almost completely inhibit spore germination in *R. solani* and *B. cinerea* [34,35]. A GH18 chitinase obtained from a different Pseudomonas species, which shares significant amino acid sequence homology with chitinases from *Serratia marcescens*, enhanced the insecticidal potency of *Spodoptera litura* nucleopolyhedrovirus [36]. Brandt et al. [37] observed the degradation of the peritrophic membrane of *Orgyia pseudotsugata* by chitinases. This phenomenon was subsequently observed using *Spodoptera littoralis* and *E. coli* expressing the endochitinase ChiAII from *Serratia marcescens* [38]. Chitinase B obtained from *Paenibacillus illinoisensis* caused deformation and degradation of the eggshell of the root-knot nematode (*Meloidogyne incognita*) [39]. Moreover, it is worth noting that β-1,3-glucanase and chitinases have been identified as significant contributors to plant defense against pathogens. They are closely associated with the systemic acquired resistance response in plants [40].

Such findings suggest that chitinases can be appropriately utilized as green insecticides to prevent and control *S. frugiperda* in the future [33,40]. Concerns about the environmental release of transgenic bacteria led to the further improvement of the crude extract of chitinase B, which was then utilized as a pathogenic agent against *S. frugipedra*. In our study, we expressed the chitinase B heterologously in the expression host *E. coli* BL21 (DE3) and purified the protein for its inaugural use as an insecticide against *S. frugipedra*. Through heterologous expression, the yield of chitinase B increased 2.5-fold compared to the wild strain. This approach allowed us to assess the potential of chitinase B as a standalone bioagent for controlling *S. frugipedra*. Considering the economic analysis of producing and deploying chitinase B as a biocontrol agent, it becomes evident that establishing an in vivo factory connected to the transformant expression host (*E. coli* (DE3)) and utilizing these bioagents in a partially purified form through ethanol precipitation represents a highly cost-effective strategy. This stands in stark contrast to the expenses incurred by the chemical insecticide industry. In conclusion, the data presented in this study strongly support the notion that these chitinases hold great potential as both effective and economically feasible biological control agents.

Demonstrating the interaction between chitinase B and chitin could be clearly investigated through the approach of computational modeling and docking studies. However, despite the limitations compared to experimental approaches, it is essential to explore the structure–function relationship and substrate–protein interactions of proteins at a minimal cost [41]. Web-based servers validated the 3D structural model of chitinase B, relying on data calculations from Wiederstein and Sippl [42], Rishad et al. [43], and Oduselu et al. [44]. Molecular docking analysis indicated that chitin establishes favorable interactions with chitinase through the enzyme’s active site. Among these amino acids, glutamic acid, specifically one glutamic acid (Glu144) molecule, was found to bind to the substrate, chitin, through two hydrogen bonds. The validation of the docking protocol involving the receptor chitinase B and the ligand chitin was confirmed based on two key criteria: binding score and root-mean-square deviation (RMSD). A significant interaction affinity score of −5.71 kcal/mol coupled with an RMSD of 1.3 Å indicates the reliability of the chosen pose because RMSD < 2.0 Å corresponds to good docking solutions [45] along with the best scoring energy. It is worth noting that several other poses also met these criteria. However, it is strongly advised to select the best docking solution based not only on the scoring function but also on additional structural criteria outlined for similar ligands to ensure the accuracy of the selection. These docking orientations are in line with previous studies [35], who oriented the self-docking of chitinase with its co-crystallized ligand and demonstrated that the active site, comprising Asp215 and Glu144 (the catalytic acid), interacted with cyclic dipeptides that served as chitinase inhibitors. Moreover, information from a web-based server (https://www.uniprot.org/uniprotkb/P11797/entry#interaction, accessed on 22 November 2023) confirmed the representation of the active site by Glu144. Based on the available information from previous reports [35], the available structural data further reinforce the reliability of selecting the molecular docking top-scoring position as the best solution [46].

Future prospective: It appears highly rational that forthcoming studies should maintain their focus on identifying the benefits and drawbacks of employing chitinase for the management of *S. frugipedra*, and investigating the potential evolution of resistance, akin to that observed with chemical insecticides. This investigation could be adequately evaluated by applying concentrations both lower and higher than those utilized in our study in a repetitive manner and recording the response of the insects in terms of malformation, mortality, developmental alterations, and the establishment of resistance.

## 4. Materials and Methods

### 4.1. Isolation and Maintenance of Chitinase-Producing Bacteria

Twenty naturally infected *S. frugiperda* larvae were collected from farms in El-Qaliobia and Menoufia Governorates, Egypt, where synthetic insecticides were not applied (Figure 13). Larvae were transported to the laboratory under aseptic conditions in sterile plastic bags and washed with 70% ethanol and sterile tap water, followed by drying in a laminar flow hood for 10 min. Subsequently, they were minced using a sterile platinum loop, and streaked onto sterile nutrient agar in Petri dishes, which were then incubated at 30 ± 1 °C for 48 h. Bacterial colonies exhibiting similar morphologies and dominance were subsequently streaked onto nutrient agar (NA) plates under identical conditions to isolate pure bacterial cultures. Following purification, the isolates were re-cultured and stored at 4 °C on nutrient agar slants for subsequent investigation [47].

### 4.2. Chitinolytic Assay

#### 4.2.1. Preparation of Production Media

Three milliliters of the overnight bacterial culture was inoculated in a sterilized 250 mL Erlenmeyer flask containing the following components in grammes per 50 mL: shrimp shells 1, yeast 0.5, and KH2PO_4_ 0.1. This condition was assigned as the standardized condition. The temperature and shaking conditions were adjusted to 30 °C and 180 rpm, respectively, for two days. The culture was centrifuged to obtain the supernatant that was employed as an enzyme source [48].

#### 4.2.2. Preparation of Colloidal Chitin

Following the method outlined by Wang et al. [49], pure chitin (Sigma-Aldrich, Burlington, MA, USA) was used to prepare colloidal chitin. Forty grams of chitin was placed in a beaker, then 500 mL of 37% HCl (concentrated) was supplemented under continuous stirring at 4 °C. After 1 h of stirring, the hydrolyzed chitin in the beaker underwent several washes with distilled water to completely eliminate the acid, bringing the pH into the range of 6–7. Once the desired pH was achieved, the colloidal chitin was filtered using Whatman filter paper. The resulting filtered colloidal chitin was collected and stored as a paste at 4 °C.

#### 4.2.3. Assay of Chitinase Activity

Chitinase activity was assessed following the procedure outlined by Monreal and Reese [50], where the detection of N-acetylglucosamine (NAG) served as the final product. The chitinase assay’s reaction mixture consisted of 1 mL of 5% acid-swollen chitin, 1 mL of 50 mM acetate buffer (pH 5.0), and 1 mL of the supernatant as an enzyme source. The reaction mixture underwent incubation at 50 °C for 1 h, followed by termination through boiling for 15 min. Subsequently, the mixture was centrifuged at 5000 rpm for 20 min, and the concentration of released NAG was measured spectrophotometrically at 510 nm, using colloidal chitin as the substrate. Chitinase activity was defined as one unit when the enzyme catalyzed the release of 1 μmol of N-acetylglucosamine (NAG) per hour at 50 °C. The best bacterial isolate in chitinase production was assigned NRC408.

#### 4.2.4. Bacterial DNA Purification and PCR Amplification

The pure isolate NRC408—the highest chitinase producer—underwent an overnight incubation in nutrient broth (NB) at 30 °C to extract bacterial DNA. The purification of genomic DNA was accomplished through the utilization of Thermo Scientific™’s GeneJET Genomic DNA Purification Kit (Waltham, MA, USA). Universal bacterial primers were employed for amplifying the 16S rRNA gene, with the 8F forward primer (5′-AGAGTTTGATCMTGGCTCAG-3′) and the 1429R reverse primer (5′-TACGGYTACCTTGTTACGACTT-3′). The PCR reaction mixture (50 µL) included 2 µL of each primer (forward and reverse) with a concentration of 10 pmol, 25 µL of 2× EmeraldAmp^®^ GT PCR Master Mix from Takara Bio (Shiga, Japan), and 4 µL of bacterial DNA template, and was adjusted to 50 µL with PCR-grade water. PCR involved an initial denaturation at 95 °C for 3 min, followed by 35 cycles of denaturation at 95 °C for 30 s, annealing at 50 °C for 30 s, and elongation at 72 °C for 1 min. The final extension was carried out for an additional 10 min at 72 °C using the BioRad T100 Thermal Cycler. The amplified fragment was examined on a 1.2% agarose gel stained with ethidium bromide. PCR products were purified using the GeneJET Gel Extraction Kit from Thermo Scientific™ (Waltham, MA, USA) and were subsequently sent to Macrogen Inc. (Amsterdam, the Netherlands) for sequencing of the 16S rRNA using the high-throughput Applied Biosystems 3730XL sequencer.

#### 4.2.5. Phylogenetic Analysis

The sequence underwent editing through BioEdit 7.1.10 software [51]. To identify similar species, the obtained sequences were compared to those in the GenBank database (http://www.ncbi.nlm.nih.gov/blast, accessed on 22 November 2023). A set of 16 related species’ 16S rRNA genes from the GenBank database was utilized to construct a phylogenetic tree. Employing the MUSCLE [52] algorithm in MEGA11 [53], multiple sequence alignments were conducted. The evolutionary history was deduced using the neighbor-joining method [54] with a 1000-bootstrap run [55], and the Jukes–Cantor method [56] was applied to compute evolutionary distances.

### 4.3. Molecular Cloning and Heterologous Expression of ChiB Gene

#### 4.3.1. Bacterial Strains, Growth Media, and Plasmids

In this investigation, three bacterial strains were employed: the bacterial isolate NRC408 served as a DNA source for the isolation of the chiB gene, *E. coli* DH5α functioned as a surrogate host for the constructed cloning vector, and *E. coli* BL21 (DE3) was utilized for the expression and production of the recombinant chitinase. The latter two strains were provided by Invitrogen, based in San Diego, California. All bacterial strains were cultured in Lysogeny broth (LB) media [57], allowing for DNA isolation, gene cloning, and plasmid propagation. A CloneJET PCR Cloning Kit from Thermo Scientific™ (Waltham, MA, USA) was employed to clone the chiB gene and propagate it into *E. coli* DH5α. The pET-28a (+) vector from Novagen (Madison, WI, USA) was designated as the vehicle for gene expression in *E. coli* BL21 (DE3).

#### 4.3.2. Primer Design and ChiB Gene Amplification

Based on the chitinase B protein sequence documented in the Uniprot database (https://www.uniprot.org/uniprotkb/P11797/entry#sequences, accessed on 22 November 2023), the primer design was accomplished. Also, the ChiB gene sequence retrieved from EMBL (https://www.ebi.ac.uk/ena/browser/view/X15208, accessed on 22 November 2023) under the X15208.1 entry was used to accomplish this task utilizing the Primer3 program (https://bioinfo.ut.ee/primer3-0.4.0/, accessed on 22 November 2023). The nucleotide sequence of the forward primer, ChiB-F, is 5′-GCGAATTCATGTCCACACGCAAAGCCGTTA-3′, encompassing an EcoRI recognition site underlined for emphasis; and that of the reverse primer, ChiB-R, is 5′-CCAAGCTTTTACGCTACGCGGCCCAC-3′, with the HindIII recognition site underlined. New England Biolabs (NEB) supplied the restriction enzymes, and the primers were synthesized by Macrogen in Korea. PCR-based amplification of the ChiB gene was conducted using Phusion High-Fidelity DNA Polymerase from Thermo Scientific™ (Waltham, MA, USA), adhering precisely to its provided protocol. The annealing temperature employed was 60 °C.

#### 4.3.3. Gene Cloning and Transformation

The PCR product was subjected to electrophoresis on a 1.2% agarose gel, and the band containing gene fragments was cut out and processed using GeneJET Gel Extraction Thermo Scientific™ (Waltham, MA, USA). The purified ChiB gene fragment containing the ORF was then inserted into pJET1.2/blunt using the CloneJET PCR Cloning Kit Thermo Scientific™ (Waltham, MA, USA), following the provided instructions. Competent *E. coli* DH5α cells were prepared based on the procedure outlined by Chung et al. [58], and subsequently transformed with the ligation product. To identify successful transformants, the cells were allowed to grow under carbenicillin selection on LB agar plates with a concentration of 50 µg/mL for 18 h at 37 °C. Positive transformants were identified by the presence of growing colonies, as the vector contains a lethal restriction enzyme gene that is interrupted by the ligation of a DNA insert into the cloning site. Consequently, only bacterial cells with recombinant plasmids could form colonies. The resulting constructed vector was designated as pJET-ChiB.

#### 4.3.4. Heterologous Expression of Chitinase B in *E. coli* BL21 (DE3)

The engineered vector, pJET-ChiB, was obtained from transformed cells using the GeneJET Plasmid Miniprep Kit from Thermo Scientific™ (Waltham, MA, USA) following the provided instructions. Subsequently, a concurrent double digestion of both pJET-aprE and pET-28a (+) vectors was performed using EcoR1 and HindIII. Following digestion, the obtained products were resolved on agarose gel—a concentration of 1.2%. Two bands, corresponding to the gene fragment and pJET, were isolated via the GeneJET Gel Extraction Kit (Thermo Scientific™) and then ligated using T4 DNA Ligase (Thermo Scientific™) in accordance with the recommended procedures. At this stage, the ligated product was ready for introduction into the competent cells of *E. coli* DH5α, which were subsequently cultivated under kanamycin selection (50 µg/mL in LB agar media). All gene cloning and transformation steps adhered to the protocols outlined by Sambrook and Russell [59]. Transformant cells capable of growth under kanamycin selection were selected and propagated in LB broth containing 50 µg/mL kanamycin. Verification of the positive transformant containing ChiB-pET28a clones and the integrity of the recombinant plasmid was conducted using two methods: colony PCR with a set of primers for the target gene and pET-28a (+) T7 promoter primers, along with double digestion of the recombinant vector using EcoRI and HindIII. Following confirmation of the correct conformation of the recombinant vector, it was isolated and introduced into *E. coli* BL21 (DE3) competent cells, with cultivation under kanamycin selection.

#### 4.3.5. Expression of Chitinase B in *E. coli* and SDS-PAGE

The positive transformant, which harbors the pET-ChiB construct, was cultivated on LB agar media supplemented with kanamycin (50 µg/mL). A singular colony from a clone was utilized to inoculate 5 mL of LB broth with kanamycin (50 µg/mL), followed by incubation at 37 °C in a shaking incubator. One milliliter of the overnight culture was employed to inoculate 200 mL of LB broth with kanamycin (50 µg/mL) in a 500 mL culture flask, and the culture was incubated at 37 °C with vigorous shaking. Once the cell density reached a mid-log phase of 0.6 at OD600, isopropyl β-D-1-thiogalactoside (IPTG) was added at different concentrations, 0.1, 0.2, 0.3, 0.4, and 0.5 mM, and different incubation temperatures, 18, 28, and 37 °C, and the culture underwent additional shaking for 16 h after induction [60]. Subsequently, the cells were harvested, rinsed with deionized water, and suspended in 10 mmol/L PBS at pH 7. The cells underwent sonication for 10 min, with cycles of 30 s of sonication followed by a 30 s interval, while being kept chilled on ice. The resulting cell lysate was then centrifuged for 10 min. Chitinase protein was precipitated according to previously described methods [61,62]. Briefly, a volume of approximately 250 mL of lysate was subjected to fractional precipitation at 4 °C using cooled ethanol at varying concentrations of 0–20%, 20–40%, 40–60%, 60–70%, and 70–90%. Ethanol was gradually added to the supernatant, and the resulting protein precipitate was obtained by centrifugation at 8000× *g* for 15 min. Subsequently, the precipitate was dialyzed against 50 mM potassium phosphate buffer at pH 7 for 12 h at 4 °C. Following dialysis, the chitinase activity and protein content of each fraction were determined using Bradford reagent (BioRad, Hercules, CA, USA). Following the Laemmli method [63], Sodium Dodecyl Sulfate-Polyacrylamide Gel Electrophoresis (12% SDS-PAGE) was employed to analyze the purified target protein. Coomassie Brilliant Blue R-250 was used for visual band detection, and a pre-stained protein ladder (Thermo Scientific™) aided in identifying the molecular weight of the protease.

#### 4.3.6. DNA Sequencing and Analysis of ChiB Gene

The vector primers for pET28a (+), namely pET-F (5′-CGTCCGGCGTAGAGGATC-3′) and pET-R (5′-ATCCGGATATAGTTCCTCCTTTC-3′), were employed to determine the nucleotide sequence through the dideoxynucleotide chain termination method [64] using BigDye terminator cycle sequencing ready reaction kits from Macrogen Inc. (Amsterdam, the Netherlands). Subsequent to sequencing, editing was performed to rectify inaccuracies and trimming was performed to eliminate unreadable portions at the 3′ and 5′ ends. BioEdit version 7.0.2 software was used for these processes. The edited sequence was aligned with the NCBI nucleotide database (https://blast.ncbi.nlm.nih.gov, accessed on 22 November 2023) and assigned a specific accession number. The deduced amino acids of the ChiB gene were aligned with the protein database in Uniprot (https://legacy.uniprot.org/align/, accessed on 22 November 2023). In order to validate the accuracy of our sequence and structural alignment, and to delineate the evolutionary relationships and degree of homology among GH-18 chitinases, which include chitinase B, we retrieved seven representative structures of family 18 chitinases (Q9REI6, 1E15, Q9REI6, Q873X9, Q13231, Q9BZP6, and P0CB51) from GenBank. This retrieval was performed by querying the protein database using the PSI-BLAST program. The retrieved records encompassed chitinases from Bacteria, Fungi, and Animalia. The structure-based alignment, inclusive of the superimposed predicted secondary structures, was executed using the ESPript 3.0 program [65]. The prediction of secondary structures involving the alpha helix and beta sheets was accomplished by utilizing the pdbsum web server (https://www.ebi.ac.uk/thornton-srv/databases/pdbsum/Generate.html, accessed on 22 November 2023). Identification of conserved domains in the deduced amino acid sequence was accomplished by searching on https://www.ncbi.nlm.nih.gov/Structure/cdd/wrpsb.cgi, accessed on 22 November 2023. Analysis of the N-terminal signal peptide was conducted using the SignalP version 6.0 program (http://www.cbs.dtu.dk/services/SignalP/, accessed on 22 November 2023) [66]. The molecular mass and isoelectric point (pI) of the encoded chitinase B were determined using the ExPASy site (http://www.expasy.ch/tools/protparam.html, accessed on 22 November 2023). To establish the phylogenetic relationship between the deduced amino acids of ChiB and other proteins with high identity, MEGA11 was employed [53]. Multiple sequence alignment was performed using the MUSCLE algorithm [52]. The evolutionary history was deduced using the neighbor-joining method [54] with a 1000-replicate bootstrap test [55].

#### 4.3.7. Template Search and Homology Modeling

The nucleotide sequence translation and alignment of deduced amino acid sequences were carried out utilizing the Expasy translate tools available at web.expasy.org/translate/. To construct the homology model, SWISS-MODEL [67,68,69,70] was employed. The input for the protein sequence was derived from the translation of the chitinase B encoded by the ChiB gene in NRC408. To develop the model, a template search using Blast and HHBlits was executed on the SWISS-MODEL web server, ensuring adequate coverage of the query sequence and sequence identity within the template library. The BLAST tool [71] was employed to search the target sequence against the primary amino acid sequence in the SMTL. The evaluation of both Global Model Quality Estimation (GMQE) [58] and Qualitative Model Energy Analysis (QMEAN) values [59] was necessary in order to select the most reliable 3D structure. Higher numbers indicate more reliability of the projected structure, and GMQE values of 10 lie within the 0 to 1 range. Reliability in QMEAN is indicated by a number smaller than 4.0 [32].

#### 4.3.8. Structure Validation of Modeled Protein

The authentication of the modeled protein structure was conducted using the ProSA server (https://prosa.services.came.sbg.ac.at/prosa.php, accessed on 22 November 2023) and SAVES v6.0 (Structure Analysis and Verification Server version 6), which test the atomic coordinates of the structure—the 3D model—to assess its structural quality. SAVES v6.0 comprises five programs designed to assess the general coherence of a protein’s structural arrangement. Among these programs, PROCHECK [60], VERIFY-3D [61], and ERRAT [62] scores were employed to evaluate the 3D protein models, while PROCHECK including Ramachandran plots and Ramachandran statistics was employed to evaluate the model’s quality by examining the allowed and disallowed regions on the plot [44]. The ProSA web server generated a Z-score value, providing an indication of the overall quality of the model and its similarity to native nuclear magnetic resonance/X-ray crystal structures [43].

#### 4.3.9. Alignment of the Chitinase Model and the Template Structure

The chitinase model and the template structure were aligned using the PyMOL molecular viewer [63] to visually illustrate the proximity of the carbon atoms. The root-mean-square deviation (RMSD), which measures the variation in carbon atom positions between the template and the model created by alignment, is the basis for this comparison. A lower RMSD, approaching zero, indicates a higher degree of structural similarity between the two entities. PyMOL facilitates the visualization of protein PDBs, elimination of water and heteroatoms from protein PDB files, and examination of docking outcomes. Analysis of the alpha helix, beta sheets, interacting residues, and active site has been explored using it [64].

#### 4.3.10. Molecular Docking

The computational analysis involved investigating the molecular interaction between crab-extracted chitin (N-Acetyl-beta-D-glucosamine: PubChem CID 24139) and a modeled protein using Molecular Operating Environment software (MOE) Version 2015.10 [65]. The process involved organizing, optimizing, and docking the ligand, including the removal of water molecules and addition of hydrogen atoms from the proteins. The 3D structure of the modeled protein bound with chitin was utilized for docking simulations, with protonation and energy minimization using MOE default parameters, saved in a molecular database (MDB) file [66]. The docked poses were systematically scored and rescored, and target molecule and receptor configurations were selected based on their scores and root-mean-square deviation values. The primary objective of these docking investigations was to explore the binding mechanism between the chitin (substrate) and the three-dimensional model of chitinase B. Chimera [67] was employed to visualize the receptor binding site and analyze interactions between ligands and receptors. This involved visualizing the receptor binding site, examining the arrangement of the original (co-crystallized) ligand, and scrutinizing the primary ligand–receptor interactions, particularly those involving hydrogen bonding with crucial amino acid residues. The evaluation of interactions adhered to the fundamental criteria outlined in a Medicinal Chemist’s Guide to Molecular Interactions [68].

### 4.4. Insecticidal Activity Assessment

#### 4.4.1. Laboratory Screening

From the precipitated protein at an ethanol fraction of 60–70%, approximately three and a half milligrams were dissolved in one milliliter of phosphate buffer, pH 7, obtaining a concentration of 2000 U/mL. Maize leaves, each measuring 20 mm in diameter, were immersed in the enzyme preparation for 5 min. After air-drying, fourth instar larvae were fed with enzyme-coated leaves following an 8 h starvation period. Each experimental group comprised 100 larvae divided into four groups (four replicates) maintained in 12:12 light/dark photoperiods at 30 °C with 60% RH, and experiments were conducted in triplicate, with daily mortality recordings over a 15-day period. The same diet, soaked in distilled water, was introduced to control groups. The recorded data were documented and computed following the methodology of Herlinda et al. [72] and based on previous research by Ayudya et al. [73]. Larvae were considered dead if they did not move or react when touched with a camel hairbrush. Larval mortality values were corrected using Abbott’s formula. Emamectin benzoate (GT Biochemical Co., Nanning, China) was used as a positive control, following Liu et al. [74]. Evaluation of mortality and biological aspects including growth and development of *S. frugiperda* larvae following exposure to recombinant chitinase B was performed.

#### 4.4.2. Field Evaluation

Field trials were carried out in El-kilo 56 district, Behera Governorate, during 2022 (June to October) and 2023 (July to November). The experimental area was 378 m^2^, divided into nine plots; each plot was 42 m^2^, and was divided into three completely randomized blocks. The maize seeds planted in each replicate were regularly irrigated and were supplied with recommended doses of fertilizers to maintain good plant health until harvest. The commercial insecticide Emamectin benzoate (1-Spedo^®^ 5.7%WG, name of manufacturing company) was used as a positive control, at a rate of 80 g/fedan as a recommended rate. A crude enzyme solution with a concentration of 2000 U/mL was obtained, and each plant was sprayed with 10 mL. During both seasons, as soon as pest incidence was observed, treated plants were sprayed by a knapsack sprayer twice, separated by ten days. Plant infestation with FAW was recorded based on the number of plants infested by the pest. Water was assigned to the control group. Thirty maize plants were selected at random to monitor FAW larvae and record the numbers before application and after the 1st, 3rd, 5th, and 10th days from application. The experimental procedure employed in this experiment closely adhered to previous study [75], with certain modifications. The reduction percentage of pest population density was calculated using the Henderson and Tilton equation [76].

#### 4.4.3. Statistical Analysis

Laboratory and field assessment results were analyzed by one-way ANOVA using SPSS version 23.0 (IBM Corp., Armonk, NY, USA, 2015). According to Tukey’s test at *p* < 0.05, different lowercase letters indicate statistical differences between treatments, but those with the same letter are not significantly different.

## 5. Conclusions

In this study, a strain of *S. marcescens* NRC408 was isolated and identified as a potential chitinase producer. The heterologous expression of ChiB in *E. coli* strain BL21 (DE3) was driven by pET28a (+) as an expression vector, resulting in a higher yield of recombinant chitinase B, which was then perfectly purified using cooled ethanol at a fraction of 60–70%. The purified recombinant enzyme was successfully used as a bioagent to control *S. frugipedra*. These results demonstrate that chitinase B shows excellent insecticidal activity against FAW larvae and negatively affects FAW development.

## Figures and Tables

**Figure 1 molecules-29-01466-f001:**
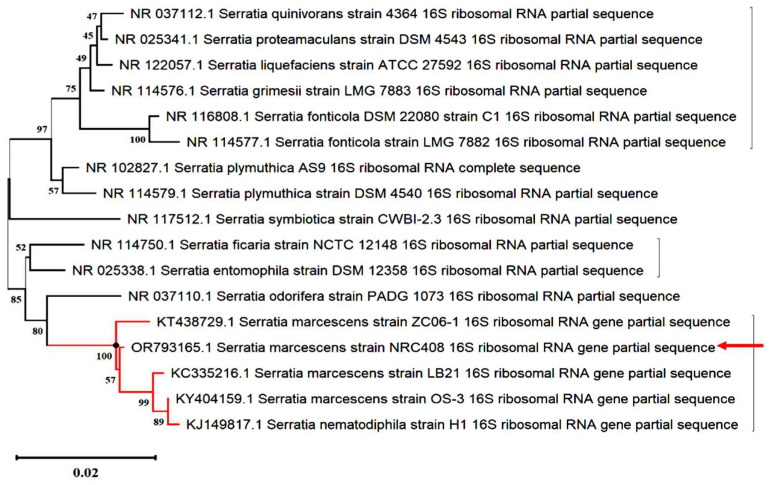
The phylogenetic relationships among bacterial isolate NRC408, denoted by a red arrow, and other relevant bacteria in the GenBank database were established using the neighbor-joining method. The numerical values at nodes represent bootstrap percentages obtained from 1000 replications.

**Figure 2 molecules-29-01466-f002:**
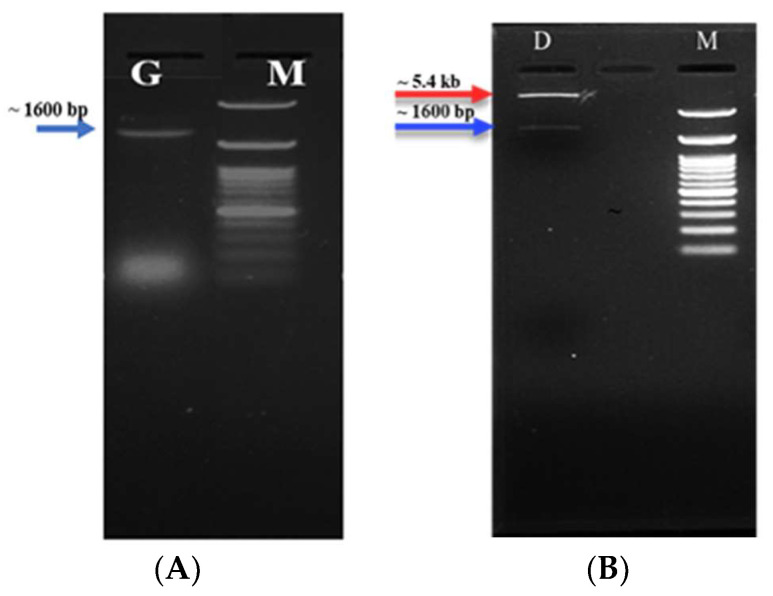
Confirmation of the correct construction of the recombinant vector, pet28-ChiB, was carried out through two methods: (**A**) colony PCR of positive transformants and (**B**) double digestion of the recombinant vector using EcoRI and HindIII restriction enzymes. G: the amplified fragment of the ChiB gene (1600 bp), D: the digested products of the recombinant vector; the red arrow refers to the pET-28a (+) vector and the blue arrow refers to the ChiB gene. M: 100 bp DNA Ladder H3 RTU (GeneDirex, Inc., Taoyuan, Taiwan).

**Figure 3 molecules-29-01466-f003:**
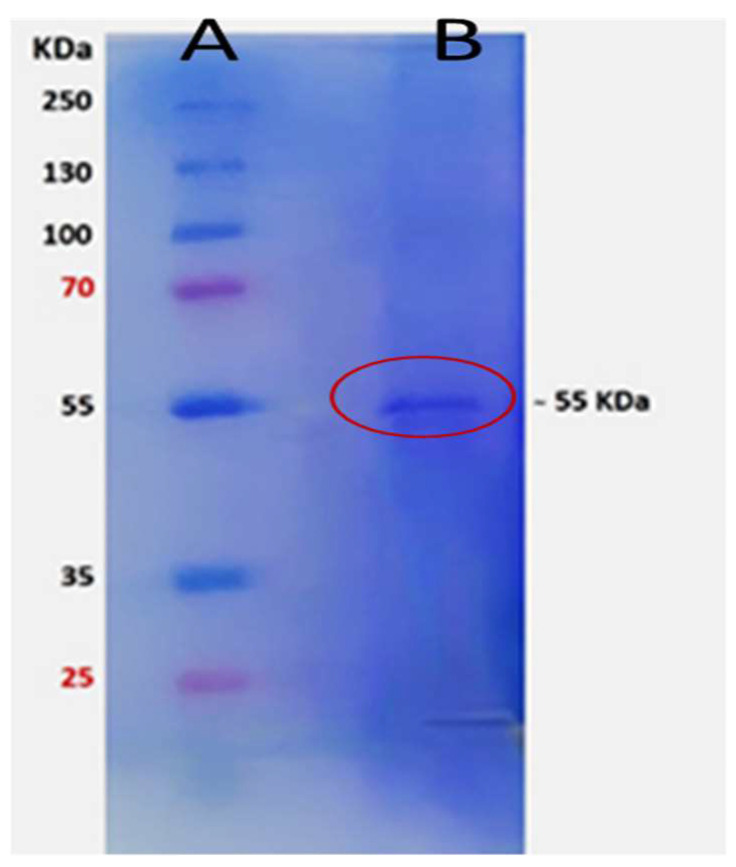
SDS-PAGE of partially purified rHis6-ChiB. Lane A: low-molecular-weight protein ladder from Thermo Scientific™, Waltham, MA, USA; Lane B: protein band with a molecular weight of 55 kDa, indicating chitinase B and marked by a red circle.

**Figure 4 molecules-29-01466-f004:**
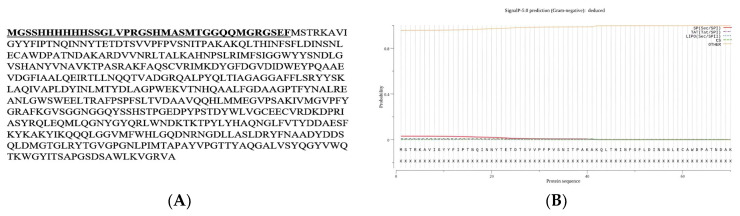
(**A**) The recombinant chitinase B’s amino acid sequence includes the N-terminal His-tag followed by the deduced amino acids of ChiB, along with additional amino acid sequences from pET28a. Among these sequences, a T7 tag is present and highlighted in bold and underlined letters. (**B**) Signal peptide and pro-peptide analysis of chitinase B with Signal P, version 6.0.

**Figure 5 molecules-29-01466-f005:**
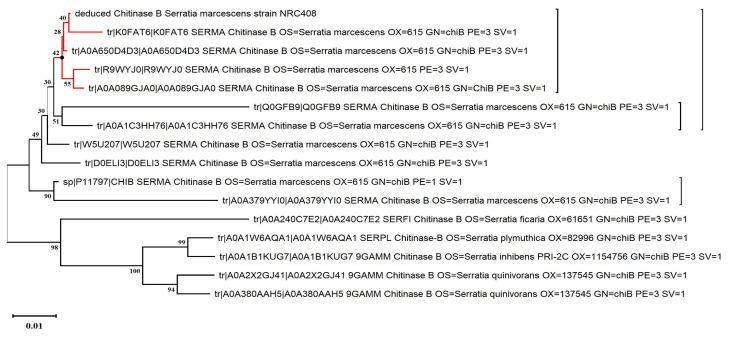
Phylogenetic tree based on a comparison of the deduced ChiB amino acid sequence (indicated by a red arrow) and its closest phylogenetic relatives retrieved from the uniprot database. The tree was reconstructed by the neighbor-joining method using MEGA 11 software. The numbers on the tree indicate the percentages of bootstrap support derived from 1000 replications.

**Figure 6 molecules-29-01466-f006:**
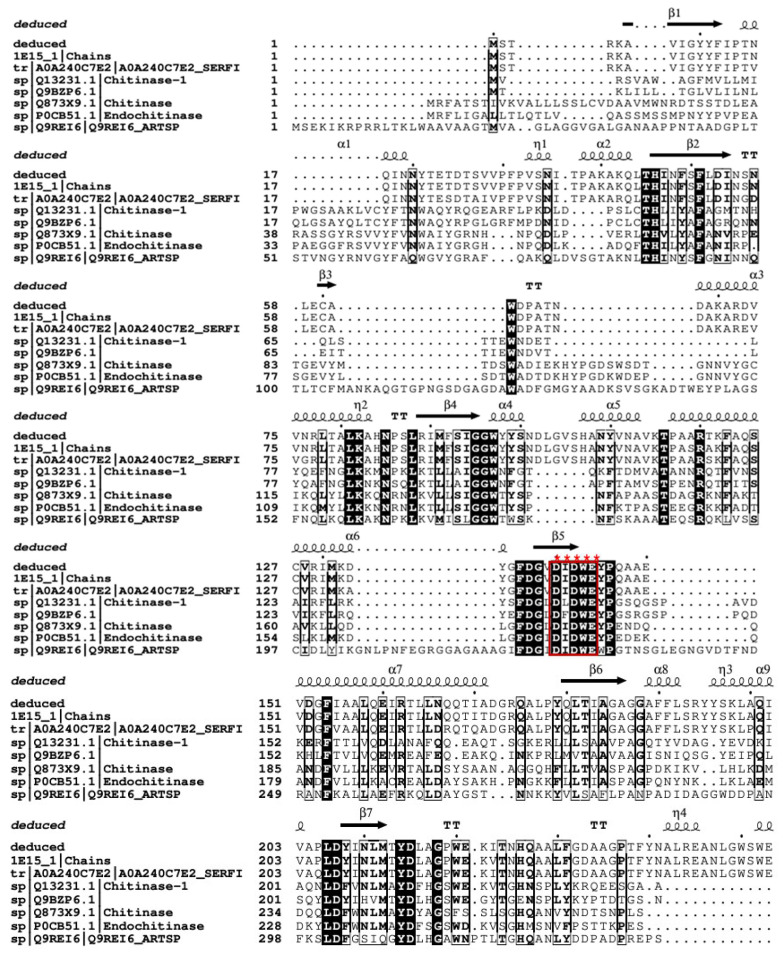
Structure-based multiple sequence alignment showing the evolutionary relatedness and homology degree among GH-18 chitinases: deduced: deduced amino acids of chitinase B, A0A240C7E2: chitinase from *Serratia ficari*, 1E15: chitinase b from *Serratia marcescens*, Q9REI6: chitinase precursor from *Arthrobacter* sp., Q873X9: endochitinase B1 from *Aspergillus fumigatus*, Q13231: chitinase-1 from Human, Q9BZP6.1: acidic precursor mammalian chitinase, P0CB51: endochitinase 1 from *Coccidioides posadasii* str. *Silveira*. The Conserved catalytic motif DXDXE is marked with red stars and outlined by red box.

**Figure 7 molecules-29-01466-f007:**
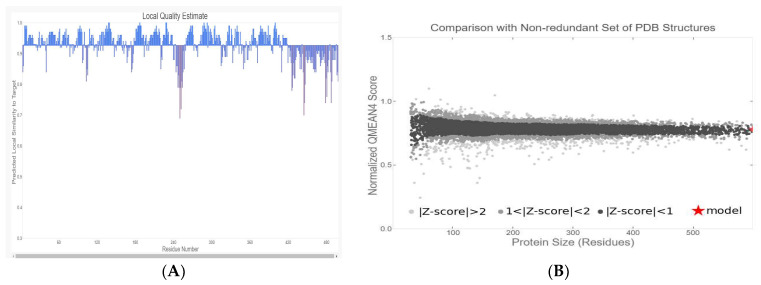
Validation of the modeled chitinase B structure involves the following: (**A**) assessing the local quality of residues in the predicted chitinase B model; (**B**) contrasting the structure of the predicted chitinase B with a nonredundant collection of PDB structures.

**Figure 8 molecules-29-01466-f008:**
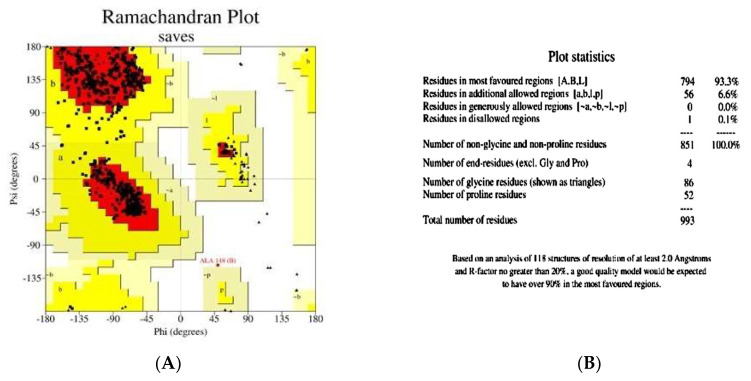
Predicted model with high score in the (**A**) Ramachandran plot analysis; (**B**) Ramachandran plot statistics of the homology model of chitinase B; (**C**) Verify3D; and (**D**) Z-score of ProSA-web.

**Figure 9 molecules-29-01466-f009:**
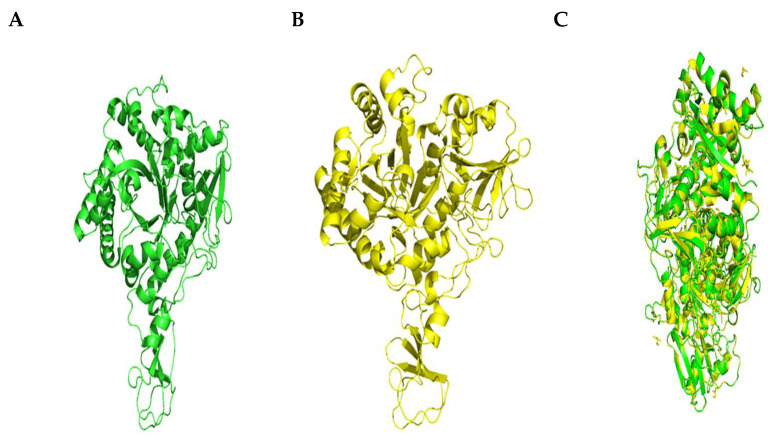
(**A**) Three-dimensional structure of the template (1W1V) model; (**B**) three-dimensional structure of homology model of chitinase B; (**C**) structure–structure alignment between target protein and template protein 1W1V.

**Figure 10 molecules-29-01466-f010:**
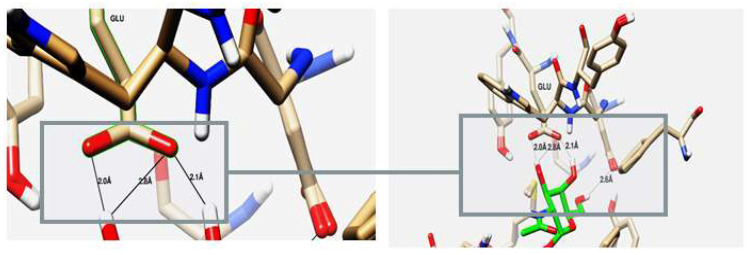
Binding disposition and ligand–receptor interactions of chitin—carbon skeleton is depicted in green—inside the chitinase B binding site, visualized with UCSF chimera software 1.17.3.

**Figure 11 molecules-29-01466-f011:**
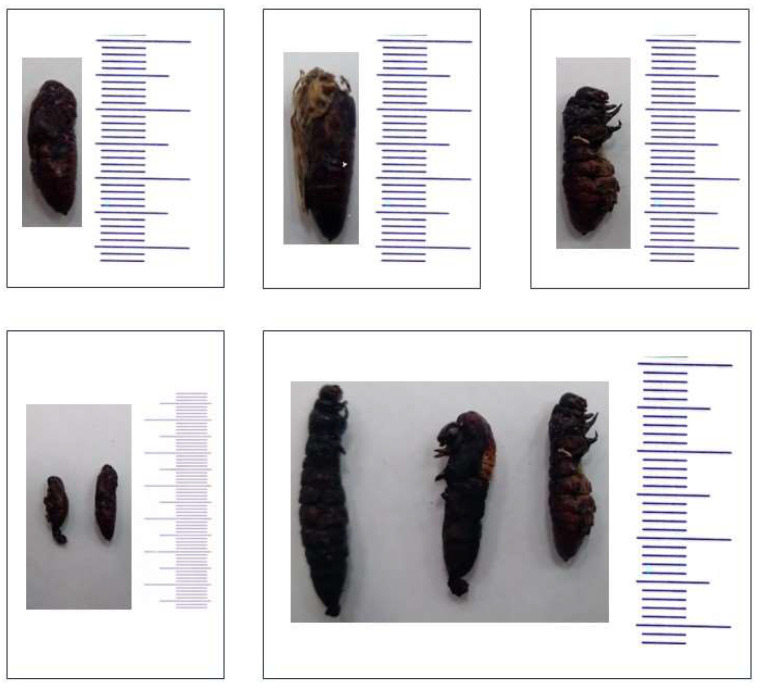
Representative malformations of *S. frugiperda* treated with chitinase B including pupal and pupal adult intermediate deformities.

**Figure 12 molecules-29-01466-f012:**
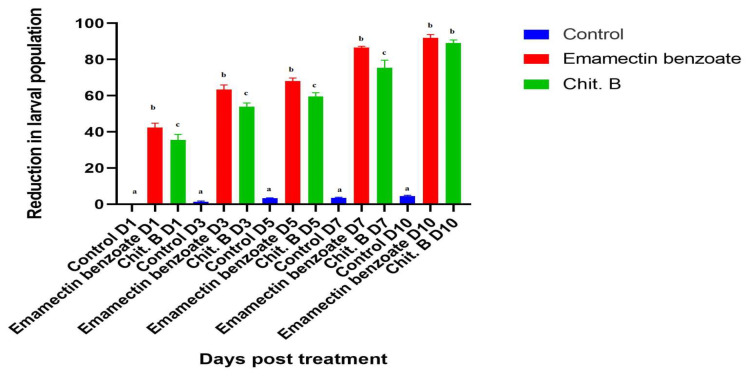
Reduction in larval pupation of *S. frugiperda* after spraying with chitinase B on maize crop. Data represented as mean and SD. Different lowercase letters indicate significant differences in biological aspects of *S. frugiperda* with chitinase B and emamectin benzoate.

**Figure 13 molecules-29-01466-f013:**
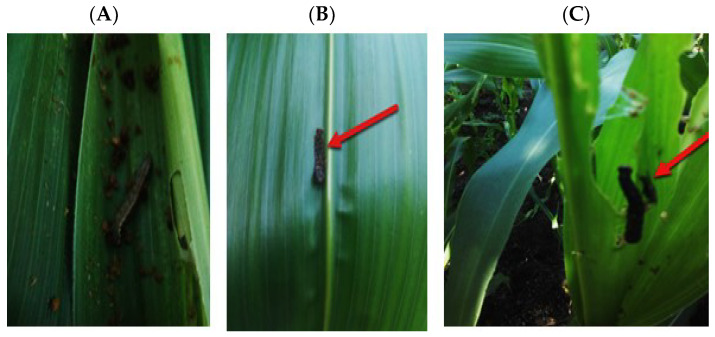
Infested corn fields with *S. frugipedra*. (**A**): healthy larvae, (**B**,**C**): infected larvae from which the pathogenic bacteria, *Serratia marcescens*, were isolated.

**Table 1 molecules-29-01466-t001:** Ethanol fractionation profile of recombinant chitinase B.

Ethanol Conc. (%)	Total Protein Content (mg)	Total Chitinase Activity (U)	Chitinase Specific Activity (U/mg)
Crude extract	20.2 ± 0.9	204 ± 1.2	10.09 ± 1.1
0–20	16.4 ± 0.6	322 ± 0.9	19.6 ± 0.6
20–40	12.4 ± 0.7	551 ± 0.9	44.4 ± 0.5
40–60	9.9 ± 0.4	905 ± 1.2	91.4 ± 0.9
60–70	4.1 ± 0.03	2581 ± 2.1	629.5 ± 1.1
70–90	3.6 ± 0.5	706 ± 1.2	196.1 ± 0.6

**Table 2 molecules-29-01466-t002:** Molecular docking interaction of chitin with *S. marcescens* chitinase B active site.

3D Representation and Energy-Minimized Structures of Chitin	2D Docking	3D Docking
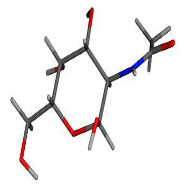	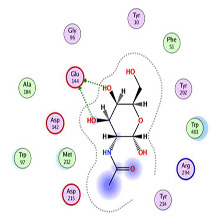	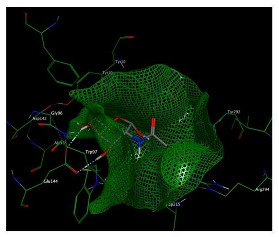

**Table 3 molecules-29-01466-t003:** Overview of the interactions between the ligand (chitin) and the binding site of the receptor (chitinase B). Hydrogen bonds are represented by a dashed bond.

Molecular Target	Ligand	Hydrogen Bond Analysis		
		Number	Hydrogen Bond Ligand/Receptor	Distance (Å)
Chitinase B	Chitin (compound CID:6857375)	2	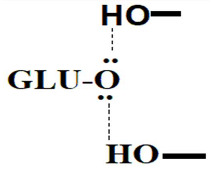	2.1 2.0

**Table 4 molecules-29-01466-t004:** Effects of chitinase B on biological aspects of *S. frugiperda*.

Biological Aspects	Control	Emamectin Benzoate	Chitinase B
Larval mortality (%)	8 ± 0.14 ^a^	98.31 ± 0.28 ^b^	92.75 ± 0.17 ^c^
Larval weight (mg)	402 ± 0.19 ^a^	278.22 ± 0.55 ^b^	296.51 ± 0.31 ^c^
Percent pupation (%)	90.83 ± 0.23 ^a^	42.9 ± 0.11 ^b^	48.92 ± 0.06 ^c^
Pupal period (days)	11.35 ± 0.36 ^a^	13.45 ± 0.09 ^b^	16.31 ± 0.25 ^c^
Pupal weight (mg)	150.02 ± 0.09 ^a^	122.14 ± 0.15 ^b^	130.23 ± 0.87 ^c^
Eclosion rate (%)	88.16 ± 0.41 ^a^	62.52 ± 0.27 ^b^	68.41 ± 0.19 ^c^
Adult longevity (days)	10.71 ± 0.015 ^a^	13.21 ± 0.03 ^b^	16.37 ± 0.72 ^c^
Hatching rate (%)	93.01 ± 0.62 ^a^	55.81 ± 0.42 ^b^	58.14 ± 0.64 ^c^

Data represented as mean and SD. Different lowercase letters indicate significant differences in biological aspects of *S. frugiperda* with chitinase B and emamectin benzoate.

## Data Availability

All data are available in the manuscript.

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
