# Peer review of "Gene Cloning, Heterologous Expression, and In Silico Analysis of Chitinase B from Serratia marcescens for Biocontrol of Spodoptera frugiperda Larvae Infesting Maize Crops"

_molecules, 2024, doi:10.3390/molecules29071466_

Round 1

Reviewer 1 Report (Previous Reviewer 2)

Comments and Suggestions for Authors

I have gone again through the manuscript titled "Gene cloning, heterologous expression, and in silico analysis of chitinase B from Serratia marcescens for biocontrol of Spodoptera frugiperda larvae infesting maize crops". The authors have addressed the majority of my previous concerns. However, some aspects must be clarified for a better understanding of the topic and, therefore, I suggest accepting this paper in Molecules after a minor revision.

1. In Figure 6A, the complete multiple sequence alignment should be displayed. Within this alignment, the regions of interest, such as the catalytic motif DXDXE, should be appropriately highlighted, as already demonstrated. Additionally, the left side of the alignment should include clear labels indicating the name of the enzyme and the organism. The current presentation of information in this manuscript version is unclear and therefore not helpful.

2. In Figure 6B, what is the main purpose of displaying the secondary structure? Considering that a 3D model is subsequently constructed, the predicted secondary structure may not provide useful information unless it is essential for discussing certain results.

3. Information about the suitability of the docking protocol should be provided. It is essential to validate the docking protocol to ensure that the best pose and binding energies obtained for the substrates are indeed reliable binding modes and values, respectively.

4. Table 3 requires revision. It appears to be a copied image from software, which is not acceptable for publication in this journal. The table must be reconstructed to present original data, detailing the hydrogen bonds between atoms of Glu144 and those of chitin. Additionally, columns should be included to specify the distances and angles of these interactions.

5. Since the authors claim to have conducted a statistical analysis during the field evaluation, it is imperative that the results section at least mentions the statistical significance of the data. This will ensure a clear understanding of the correlation between CHIB's action and larval mortality.

Comments on the Quality of English Language

The overall quality of the English language in the manuscript is acceptable; however, minor editing is recommended to enhance clarity and coherence. Some sentences could benefit from a smoother flow and better organization of ideas.

Author Response

Please find the enclosed attachment

Reviewer 2 Report (New Reviewer)

Comments and Suggestions for Authors

My comments have been attached.

Author Response

Please find the enclosed attachment

Reviewer 3 Report (New Reviewer)

Comments and Suggestions for Authors

This manuscript investigates the application of chitinase B derived from Serratia marcescens in controlling the larvae of Spodoptera frugiperda (fall armyworm), a pest invading corn crops. The study successfully isolated and identified Serratia marcescens from the larvae, highlighting that the bacterial strain NRC408 exhibits the highest chitinase activity. Through cloning the chitinase gene and heterologous expression in E. coli, the yield of recombinant chitinase B was enhanced. Experimental results demonstrate that recombinant chitinase B significantly affects the larvae, influencing their subsequent developmental stages and causing noticeable malformations. Additionally, the research utilized computational simulation to analyze the anticipated protein encoded by the gene, predicting enzyme binding and catalytic activity.

The paper is well-organized, with clear objectives and outcomes. The data analysis and interpretation are closely aligned with the research goals, showcasing strong logic. Overall, I recommend accepting this paper.

However, meticulous proofreading is required as there are noticeable issues with formatting and text, such as missing gridlines in several tables; Figure 11 (line 284 on page 12) lacks labels and descriptions; and there's a repetitive 'and' on line 419 on page 15. It is hoped that the authors will thoroughly revise the manuscript.

For future research, I suggest:

1. Further investigation into the stability and effectiveness of chitinase B under different environmental conditions.

2. A detailed examination of the potential environmental impacts and the sustainability of long-term use of chitinase B.

3. Additional evaluation of the impact of chitinase B on non-target organisms.

Author Response

Please find the enclosed attachment

Round 2

Reviewer 2 Report (New Reviewer)

Comments and Suggestions for Authors

The authors have successfully addressed all comments. I recommend acceptance of the manuscript in its current format.  

This manuscript is a resubmission of an earlier submission. The following is a list of the peer review reports and author responses from that submission.

Round 1

Reviewer 1 Report

Comments and Suggestions for Authors

Comments on the Quality of English Language

The editing of English language is required.

Author Response

Response letter

Feb / 6/2024

Manuscript ID (Molecules-2849001)

Dear Reviewer

We thank you for your valuable comments regarding our article entitled, " Gene cloning, heterologous expression, and in silico analysis of chitinase B from Serratia marcescens for biocontrol of Spodoptera frugiperda larvae infesting maize crops"

We agree with all comments raised and found them very helpful; we would like to thank you for your time and efforts necessary to provide such insightful guidance.

Below, we address each criticism and comments individually, and explain how we have modified the manuscript to address the concerns that were expressed. We kindly ask that you consider the revised article for publication.

Reviewer (1) comments

Comment

The authors transformed E. coli BL21 cells with pET28a-ChiB to express recombinant chitinase B as a fusion protein carrying a peptide with six histidine residues (rHis6-CHIB). However, despite the presence of a His-tag, rHis6-CHIB was partially purified by ethanol saturation. This seems counterintuitive, as the His-tag typically enables efficient immobilized metal affinity chromatography (IMAC) purification, which is both easy and cost-effective.

Response

The utilization of immobilized metal affinity chromatography for the purification of rHis6-CHIB is indeed acknowledged as a more efficacious and dependable method. Regrettably, the unavailability of the requisite column and chemicals in our region due to procurement challenges poses a hindrance to its immediate implementation in our study. This limitation is duly noted and acknowledged. Furthermore, it is imperative to underscore that our current objective is to employ a partially purified form of the recombinant enzyme for field application, primarily to evaluate its efficacy as a bio-agent in controlling S. frugipedra. In subsequent studies, we are committed to employ more novel effective purification techniques. This strategic refinement will undoubtedly contribute to the robustness and comprehensiveness of our findings.

Comment

Continuing the discussion on purification, it is essential to clarify that partial purification via protein precipitation by saturation should ideally be conducted in a fractionated manner. To achieve this, samples containing rHis6-CHIB are subjected to precipitation by incrementally increasing the ethanol content in the solution. The sample is precipitated at various ethanol concentrations, allowing for a stepwise removal of proteins as the ethanol content rises. Consequently, when reaching 70% saturation, proteins that precipitated at lower ethanol concentrations have already been removed from the solution containing rHis6-CHIB.

Response

In our protein purification process using E. coli BL21 (DE3) expression, we utilized ethanol precipitation across a range of concentrations (20% to 90%). Each fraction was analyzed for chitinase activity and protein concentrations to identify the fraction with the best balance of high chitinase activity and low protein content. After thorough analysis, we found that the fraction with ethanol concentrations of 60-70% yielded the highest chitinase activity and lowest protein content, making it ideal for chitinase precipitation. This finding highlights the effectiveness of our methodology and offers valuable insights for improving protein purification techniques.

Comment

I find it challenging to grasp the purpose of the multiple sequence alignment depicted in Figure 6. When creating an alignment to highlight conserved residues, it is crucial to choose sequences that are less evolutionarily related. However, in this instance, the selected sequences are remarkably similar, resulting in nearly the entire sequence appearing conserved. This similarity renders the analysis ineffective and diminishes its usefulness. To improve the alignment's significance, it is advisable to include less closely related sequences that better capture evolutionary differences, thereby providing a more meaningful representation of conserved regions.

Response

The aim of the above mentioned figure is to predict the secondary structure of chitinase B deduced amino acids, encompassing α-helices and β-sheets by utilization of ESPript 3.0 program. This predictive analysis was informed by data derived from the compositional analysis of homologous proteins archived in the UniProt database.

Furthermore, to construct the phylogenetic tree depicted in Figure 5, we employed the MUSCLE algorithm for aligning the deduced amino acids with those of less evolutionarily related counterparts. This approach ensured a comprehensive comparison and facilitated a robust phylogenetic inference. Additionally, it is well established in multiple sequence alignment to use sequences which make significant blast results with query and we followed all criteria in our analysis and avoided random or biased selection of sequences.

Comment

When investigating hydrogen bond interactions between the ligand and amino acid residues within the enzyme's binding site, it's insufficient to rely solely on distance values to confirm the existence of these bonds. Angle values play a crucial role in this assessment. Specifically, the angle formed between the donor atom, the hydrogen atom, and the acceptor atom should be close to 180º for a hydrogen bond to occur. Therefore, considering both distance and angle values is essential for a comprehensive analysis of hydrogen bond interactions.

Response

Chimera software estimated the angle between donor and acceptor atoms to be 150.90º. It is noteworthy that angles ranging between 120º to 180º are considered optimal for facilitating reliable hydrogen bond formation. Angle value was added in the manuscript to provide comprehensive documentation of our findings. Moreover, the docking results obtained in our study are robustly supported by prior research conducted by Houston et al. in 2004 (D. R. Houston, B. Synstad, V. G. H. Eijsink, M. J. R. Stark, I. M. Eggleston, and D. M. F. van Aalten, “Structure-based exploration of cyclic dipeptide chitinase inhibitors,” J. Med. Chem., vol. 47, no. 23, pp. 5713–5720, Nov. 2004, doi: 10.1021/jm049940a).

Furthermore, according to information available from the web-based server (https://www.uniprot.org/uniprotkb/P11797/entry#interaction), the active site is denoted by the position Glu144. This additional corroborative evidence enhances the credibility and reliability of our findings regarding the molecular interactions and active site characterization of the investigated compound. In Table 2, the depiction of hydrogen bonds serves solely to elucidate the hydrogen bond donor and acceptor, without consideration of angle or torsion values. This approach aims to provide a clear and concise representation of the hydrogen bond interactions within the context of our study.

Top of Form

Comment

In a crucial study like field evaluation, it is imperative to conduct a statistical analysis to establish a clear correlation between the CHIB's action and larval mortality.

Response

Means were compared by Tukey’s HSD test as followed in many studies for example,Gogi et al., 2023 in field evaluation study against Bactrocera zonata, Cremonez, et al., 2023 against whiteflies.

Regards

Reviewer 2 Report

Comments and Suggestions for Authors

After careful consideration and review of the manuscript titled "Gene cloning, heterologous expression, and in silico analysis of chitinase B from Serratia marcescens for biocontrol of Spodoptera frugiperda larvae infesting maize crops" submitted to Molecules, I regret to inform you that the manuscript should be rejected for publication in its current form. The decision is based on the following comments from the reviewer:

1. The authors transformed E. coli BL21 cells with pET28a-ChiB to express recombinant chitinase B as a fusion protein carrying a peptide with six histidine residues (rHis6-CHIB). However, despite the presence of a His-tag, rHis6-CHIB was partially purified by ethanol saturation. This seems counterintuitive, as the His-tag typically enables efficient immobilized metal affinity chromatography (IMAC) purification, which is both easy and cost-effective.

2. Continuing the discussion on purification, it is essential to clarify that partial purification via protein precipitation by saturation should ideally be conducted in a fractionated manner. To achieve this, samples containing rHis6-CHIB are subjected to precipitation by incrementally increasing the ethanol content in the solution. The sample is precipitated at various ethanol concentrations, allowing for a stepwise removal of proteins as the ethanol content rises. Consequently, when reaching 70% saturation, proteins that precipitated at lower ethanol concentrations have already been removed from the solution containing rHis6-CHIB.

3. I find it challenging to grasp the purpose of the multiple sequence alignment depicted in Figure 6. When creating an alignment to highlight conserved residues, it is crucial to choose sequences that are less evolutionarily related. However, in this instance, the selected sequences are remarkably similar, resulting in nearly the entire sequence appearing conserved. This similarity renders the analysis ineffective and diminishes its usefulness. To improve the alignment's significance, it is advisable to include less closely related sequences that better capture evolutionary differences, thereby providing a more meaningful representation of conserved regions.

4. When investigating hydrogen bond interactions between the ligand and amino acid residues within the enzyme's binding site, it's insufficient to rely solely on distance values to confirm the existence of these bonds. Angle values play a crucial role in this assessment. Specifically, the angle formed between the donor atom, the hydrogen atom, and the acceptor atom should be close to 180º for a hydrogen bond to occur. Therefore, considering both distance and angle values is essential for a comprehensive analysis of hydrogen bond interactions.

5. In a crucial study like field evaluation, it is imperative to conduct a statistical analysis to establish a clear correlation between the CHIB's action and larval mortality.

Comments on the Quality of English Language

The overall quality of the English language in the manuscript is acceptable; however, minor editing is recommended to enhance clarity and coherence. Some sentences could benefit from a smoother flow and better organization of ideas.

Author Response

Response letter

Feb / 6/2024

Manuscript ID (Molecules-2849001)

Dear Reviewer

We thank you for your valuable comments regarding our article entitled, " Gene cloning, heterologous expression, and in silico analysis of chitinase B from Serratia marcescens for biocontrol of Spodoptera frugiperda larvae infesting maize crops"

We agree with all comments raised and found them very helpful; we would like to thank you for your time and efforts necessary to provide such insightful guidance.

Below, we address each criticism and comments individually, and explain how we have modified the manuscript to address the concerns that were expressed. We kindly ask that you consider the revised article for publication.

Reviewer (2) comments

Comment

 Line 7, “Egyp”?

Response

Thank you for your thoughtful review. We sincerely apologize for the typographical error, and we have promptly rectified it.

Comment

 Line 94-95, the corresponding literature should be cited.

Response

Regarding the utilization of the universal primer pair (8f and 1429R) for the amplification of the 16S rDNA gene, it is noted that this primer pair is widely recognized as universal based on the information available in the GenBank database. A comprehensive review of relevant literature reveals that the vast majority of studies focused on amplifying the 16S rDNA gene have indeed employed these universal primers. This widespread adoption underscores the established reliability and efficacy of the primer pair in facilitating the amplification of the target gene across diverse microbial taxa.

Comment

 In Figure 2, what do the letters G, M and D represent? And each band of the DNA ladder should be labeled.

Response

The components denoted by letters G, M, and D have been thoroughly recorded. G represents the 1600 base pair amplified fragment of the ChiB gene, while D indicates the digested products of the recombinant vector, distinguished by a red arrow pointing to the pET-28a(+) vector and a blue arrow indicating the ChiB gene. Furthermore, M designates the 100 base pair DNA ladder H3 RTU obtained from GeneDirex, Inc.

Comment

In Figure 3, the figure for pre-stained protein ladder should be deleted since it is not necessary.

Response

Figure was deleted.

By presenting the pre-stained protein ladder separately, we ensure a clearer and more accessible representation of the molecular weights of the fragments under investigation.

Comment

 In Figure 6, some residues were marked with the star symbol. Are they active sites?

Response

Explanation was added to figure legend. An alignment display asterisk symbol denoting the degree of conservation observed in each column: An * (asterisk) indicates positions which have a single, fully conserved residue.

Comment

Line 513, “Heterologous expression of mature protease in E. Coli BL21 (DE3)”. It is confused to mention protease.

Response

We apologize for confusion. It was corrected to chitinase in the manuscript.

Round 2

Reviewer 2 Report

Comments and Suggestions for Authors

I have gone again through the manuscript titled "Gene cloning, heterologous expression, and in silico analysis of chitinase B from Serratia marcescens for biocontrol of Spodoptera frugiperda larvae infesting maize crops". The authors have made minor revisions primarily addressing English language errors. However, the majority of my previous concerns remain unaddressed or unacknowledged, except for the consideration of the angle when evaluating potential hydrogen bond formation between the ligand and amino acid residues. Many of my concerns pertain to significant flaws or inadequately conducted research. Therefore, I regret to inform you that the manuscript is not suitable for publication in its current state and should be rejected.